# A potent antimalarial benzoxaborole targets a *Plasmodium falciparum* cleavage and polyadenylation specificity factor homologue

Ebere Sonoiki[1,2,*], Caroline L. Ng[3,*], Marcus C.S. Lee[3,†], Denghui Guo[1], Yong-Kang Zhang[4], Yasheen Zhou[4], M.R.K. Alley[4], Vida Ahyong[5], Laura M. Sanz[6], Maria Jose Lafuente-Monasterio[6], Chen Dong[4], Patrick G. Schupp[3], Jiri Gut[1], Jenny Legac[1], Roland A. Cooper[7], Francisco-Javier Gamo[6], Joseph DeRisi[5], Yvonne R. Freund[4], David A. Fidock[3,8] & Philip J. Rosenthal[1]

Benzoxaboroles are effective against bacterial, fungal and protozoan pathogens. We report potent activity of the benzoxaborole AN3661 against *Plasmodium falciparum* laboratory-adapted strains (mean $IC_{50}$ 32 nM), Ugandan field isolates (mean *ex vivo* $IC_{50}$ 64 nM), and murine *P. berghei* and *P. falciparum* infections (day 4 $ED_{90}$ 0.34 and 0.57 mg kg$^{-1}$, respectively). Multiple *P. falciparum* lines selected *in vitro* for resistance to AN3661 harboured point mutations in *pfcpsf3*, which encodes a homologue of mammalian cleavage and polyadenylation specificity factor subunit 3 (CPSF-73 or CPSF3). CRISPR-Cas9-mediated introduction of *pfcpsf3* mutations into parental lines recapitulated AN3661 resistance. PfCPSF3 homology models placed these mutations in the active site, where AN3661 is predicted to bind. Transcripts for three trophozoite-expressed genes were lost in AN3661-treated trophozoites, which was not observed in parasites selected or engineered for AN3661 resistance. Our results identify the pre-mRNA processing factor PfCPSF3 as a promising antimalarial drug target.

[1] Department of Medicine, University of California, Box 0811, San Francisco, California 94143, USA. [2] Division of Infectious Diseases and Vaccinology, School of Public Health, University of California, 293 University Hall, 2199 Addison Street, Berkeley, California 94720-7360, USA. [3] Department of Microbiology and Immunology, Columbia University Medical Center, Room 1502 HHSC, 701 West 168th Street, New York, New York 10032, USA. [4] Anacor Pharmaceuticals, Inc., 1020 East Meadow Circle, Palo Alto, California 94303-4230, USA. [5] Department of Biochemistry and Biophysics, Howard Hughes Medical Institute, University of California, Box 2542, 1700 4th Street, 403C, San Francisco, California 94158, USA. [6] Malaria Discovery Performance Unit, Diseases of the Developing World, Tres Cantos Medicines Development Campus,GlaxoSmithKline, c/Severo Ochoa 2, Tres Cantos 28760, Spain. [7] Department of Natural Sciences and Mathematics, Dominican University of California, 50 Acacia Avenue, San Rafael, California 94901, USA. [8] Division of Infectious Diseases, Department of Medicine, Columbia University Medical Center, New York, New York 10032, USA. * These authors contributed equally to this work. † Present address: Malaria Programme, Wellcome Trust Sanger Institute, Wellcome Genome Campus, Cambridge CB10 1SA, UK. Correspondence and requests for materials should be addressed to P.J.R. (email: Philip.Rosenthal@ucsf.edu).

Despite efforts to curb transmission, malaria remains an important global infectious disease, accounting for an estimated 212 million cases and 429,000 deaths worldwide in 2015 (ref. 1). Challenges to the control and elimination of malaria include widespread drug resistance in *Plasmodium falciparum*, the most virulent human malaria parasite. As older regimens are limited by resistance, artemisinin-based combination therapies have been adopted as the standard of care for the treatment of uncomplicated falciparum malaria[1]. However, resistance to artemisinins has emerged in Southeast Asia[2,3], and resistance has been seen to most artemisinin-based combination therapy partner drugs[4]. The development of new antimalarial agents with novel modes of action is urgently needed.

Benzoxaboroles are boron-containing compounds that have shown potent activity against a wide range of infectious pathogens, including bacteria[5,6], fungi[7] and protozoans[8–12]. The highly electrophilic nature of the boron component of these compounds leads to interaction with a variety of protein targets via reversible covalent bonds[13,14], with identified targets including leucyl-tRNA synthetase (LeuRS)[6,7], phosphodiesterase 4 (PDE4)[15,16] and β-lactamase[17]. Tavaborole, the first benzoxaborole to receive FDA approval, is an inhibitor of fungal LeuRS that is used to treat onychomycosis[18]. Crisaborole, a PDE4 inhibitor, has completed phase 3 clinical trials for atopic dermatitis.

This manuscript reports the antimalarial profile and mechanism of action of another benzoxaborole, AN3661, which was identified from a screen of a benzoxaborole library against cultured *P. falciparum* asexual blood stage parasites[19,20]. AN3661 demonstrated potent antimalarial activity, and genetic and biochemical studies identified its target as a homologue of mammalian cleavage and polyadenylation specificity factor (CPSF) subunit 3.

## Results

### *In vitro* characterization of AN3661 activity.
An *in vitro* screen of a benzoxaborole library[19,20] against the multidrug-resistant *P. falciparum* W2 strain identified the 7-[(2'-carboxylic acid) ethyl] benzoxaborole AN3661 (Fig. 1a). AN3661 was active at nanomolar concentrations against *P. falciparum* laboratory strains known to be sensitive (3D7) or resistant (W2, Dd2, K1, HB3, FCR3 and TM90C2B) to standard antimalarial drugs, and it was similarly active in *ex vivo* studies of fresh Ugandan field isolates (Fig. 1b and Supplementary Table 1). There was no difference in AN3661 potency between drug-sensitive or -resistant parasite strains (Supplementary Table 2). AN3661 showed minimal cytotoxicity against mammalian cell lines, with the $CC_{50}$ 60.5 μM against Jurkat cells, and all other $CC_{50}$ values greater than the highest concentrations tested (25 μM or above; Supplementary Table 3).

When cultured with the *P. falciparum* 3D7 strain, AN3661 reduced the number of viable parasites in a time- and dose-dependent manner (Supplementary Fig. 1a). Parasites exposed to $10 \times IC_{50}$ concentrations of AN3661 reduced parasitemia with a rate of killing similar to that of pyrimethamine (Supplementary Fig. 1b). However, exposure to $30 \times IC_{50}$ increased the rate of

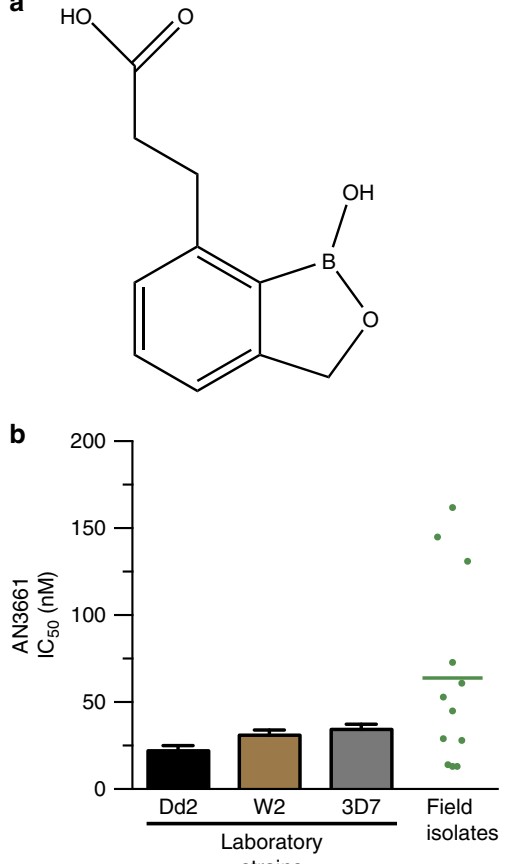

**Figure 1 | Activity of AN3661 against *P. falciparum* laboratory-adapted strains and field isolates.** (**a**) Chemical structure of AN3661. (**b**) Susceptibility of laboratory strains and field isolates to AN3661. Bar graphs represent mean ± s.e.m. $IC_{50}$ values. Each point in the dot plot represents one field isolate; the horizontal bar indicates the mean value.

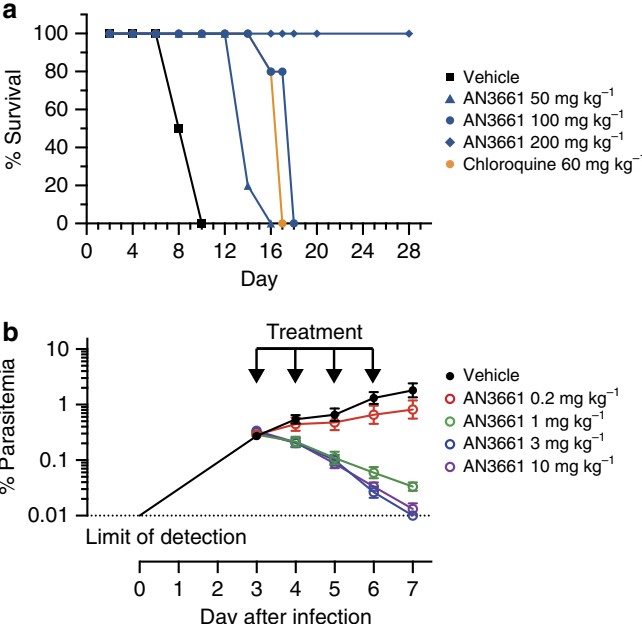

**Figure 2 | AN3661 efficacy in murine malaria models** (**a**) AN3661 efficacy in *P. berghei*-infected mice. Mice (4–5 animals for each concentration tested) were treated orally with the indicated dosages of AN3661, chloroquine or vehicle daily for 4 days. Infections were monitored daily by Giemsa-stained blood smears, and mice were euthanized when parasitemias exceeded 50%. (**b**) AN3661 efficacy in *P. falciparum*-infected mice. Mice with ∼40% circulating human erythrocytes were intravenously infected with $2 \times 10^7$ *P. falciparum*-infected erythrocytes on day 0, and AN3661 was then administered by oral gavage at the indicated dosages for 4 consecutive days (arrows). Parasitemia was measured by flow cytometry daily until day 7. Mean parasitemias for three mice in each group are shown. Error bars represent s.e.m.

killing, with a profile similar to that of chloroquine. From these results, the minimum concentration of AN3661 required to obtain the fastest kill rate was 1.38 μM.

**AN3661 activity in murine malaria models.** When administered orally for 4 days to *P. berghei*-infected mice, beginning on the day of infection, AN3661 rapidly controlled parasitemias, with a 90% effective dose ($ED_{90}$) 4 days after initiation of treatment of 0.34 mg kg$^{-1}$. Daily dosages of 50 mg kg$^{-1}$ and 100 mg kg$^{-1}$ extended survival compared to untreated controls, and mice treated with 200 mg kg$^{-1}$ per day demonstrated long-term cures (Fig. 2a). We also investigated compound efficacy against *P. falciparum* in a murine model using NODscidIL-2Rγ$^{null}$ mice engrafted with human erythrocytes and infected with *P. falciparum*[21]. When AN3661 was administered orally for 4 days, beginning on the third day of infection, the $ED_{90}$ 4 days after initiation of treatment was 0.57 mg kg$^{-1}$ (Fig. 2b).

**Stage specificity and morphology of treated parasites.** Synchronized W2 strain parasites were exposed to AN3661 for 8-h intervals across the asexual blood stage cycle, and subsequent ring-stage parasitemias were compared to those of untreated controls. Inhibitory effects of AN3661 were greatest in early to middle trophozoite-stage parasites, as found with the reference drug chloroquine (Fig. 3a). Parasites treated with 370 nM AN3661, beginning at the early ring stage, appeared morphologically normal through late rings, but could not progress beyond the trophozoite stage, with the appearance of shrunken, pyknotic parasites (Fig. 3b).

***In vitro* selection of parasites resistant to AN3661.** We quantified the ease of *in vitro* selection of resistance to AN3661 by subjecting $10^6$–$10^8$ Dd2 strain parasites to 60 nM ($2 \times IC_{90}$) AN3661. Regrowth was seen in two of three cultures with initial inocula of $10^6$ parasites at days 45 and 56, one of three cultures with initial inocula of $10^7$ parasites at day 23, and three of

three cultures with initial inocula of $10^8$ parasites, all on day 19 (Supplementary Table 4). These rates of resistance selection were similar to those observed for atovaquone.

To determine the target of AN3661, we employed two separate methods of generating resistant mutants. In the first, we pressured W2 and Dd2 strains of *P. falciparum* with increasing concentrations of AN3661 (37 nM–5 μM). In the second, a single concentration of AN3661 (170 nM) was applied to Dd2 parasites (Fig. 4a). Using the first method, for the W2 strain, 5 steps of continuous selection over 11 months led to a 200-fold decrease in sensitivity to AN3661 (Fig. 4b). After selection, W2 parasite lines with medium to high-level resistance (W2-R3, W2-R4 and W2-R5; with mean $IC_{50}$ 0.7, 6.2 and 15.3 μM, respectively, compared to 31 nM for the W2 parental line) were cultured without drug pressure to test the stability of the resistance phenotype. In each case, revertant parasites were obtained that showed decreased resistance to AN3661 after culture without the compound. For W2-R3 parasites, the revertant (W2-R3$^{rev}$) had roughly the sensitivity of the parental W2 strain, but W2-R4 and W2-R5 revertants retained micromolar $IC_{50}$ values indicating only partial loss of resistance. Results with Dd2 were similar to those with W2, with AN3661-resistant parasites obtained in 4 separate selections (mean $IC_{50}$ 0.2–0.9 μM, compared to 22 nM for the Dd2 parental line, Fig. 4c). Selection for resistance to AN3661 was accompanied by only modest changes in sensitivity to the standard antimalarial chloroquine (Supplementary Table 5).

**Whole-genome sequence analysis of AN3661-resistant parasites.** Whole-genome sequence analysis and confirmatory dideoxy sequencing of W2 and Dd2 parental and AN3661-resistant clonal lines revealed both copy number variations (CNVs) and nonsynonymous single-nucleotide polymorphisms (SNPs) in selected lines, compared to their parental lines (Supplementary Table 6). Importantly, every AN3661-resistant parasite line harboured mutations in *pfcpsf3* (PF3D7_1438500). This gene encodes a

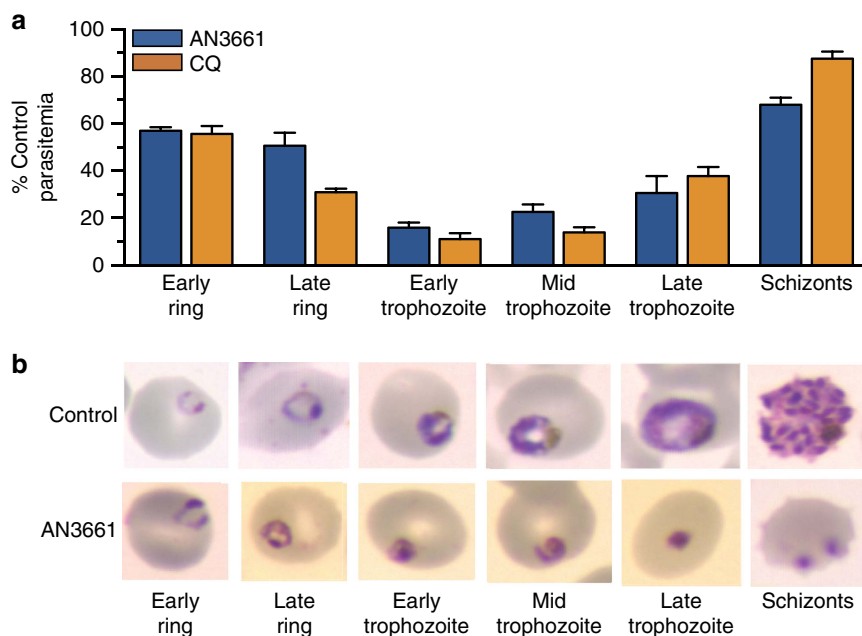

**Figure 3 | Stage specificity of AN3661.** (**a**) W2 parasites were treated with 370 nM AN3661 or 1.3 μM chloroquine (CQ) for 8-h time intervals across the asexual erythrocytic cycle. Compounds were removed at the end of each 8 h interval, cultures were continued and parasitemias were determined after 48 h. Bar graphs indicate the mean ± s.e.m. parasitemia of treated parasites as a percentage of untreated controls. (**b**) W2 parasites treated with 370 nM AN3661 beginning in the early ring stage and untreated controls were collected at the indicated time points, stained with Giemsa, and photographed.

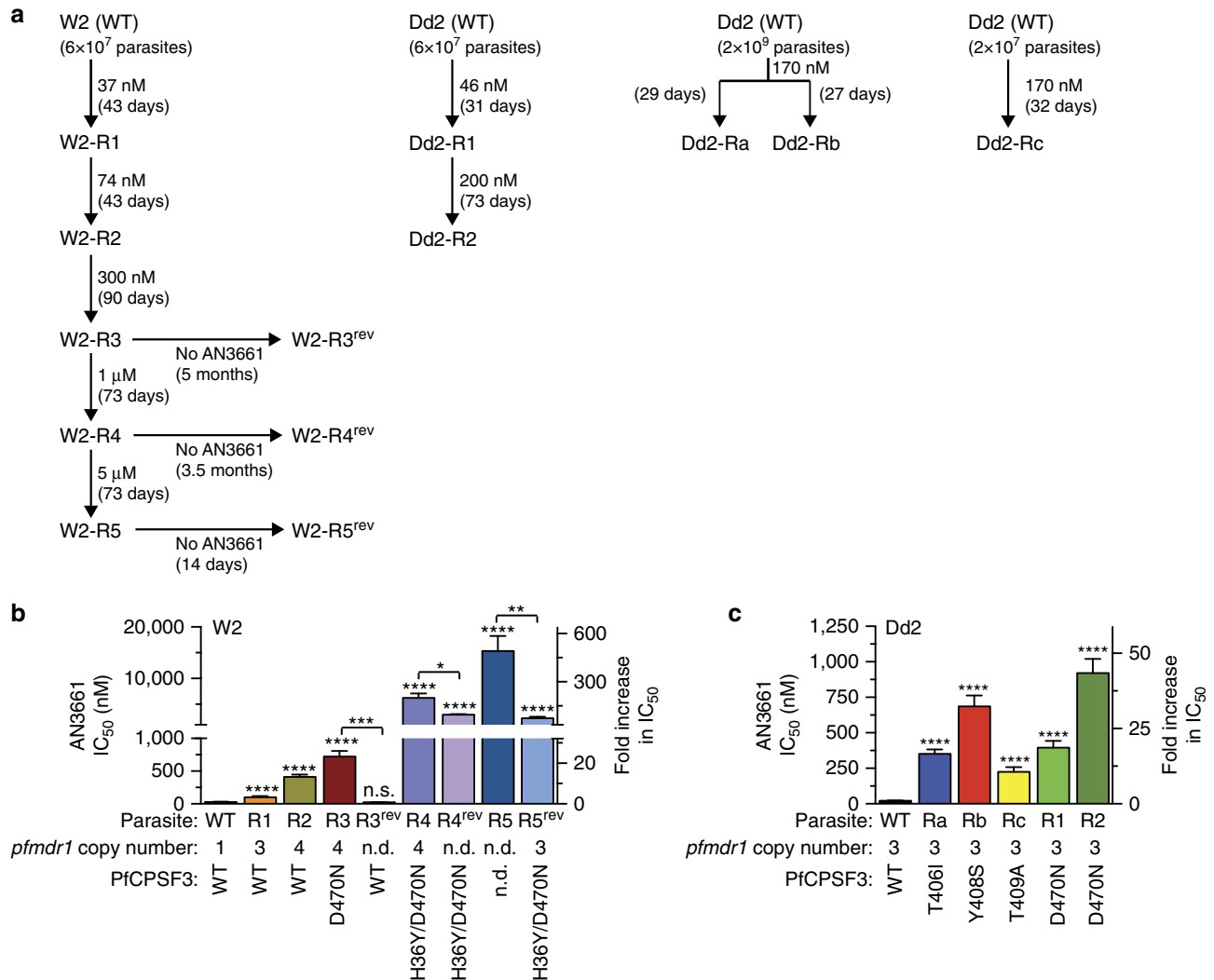

**Figure 4 | AN3661 susceptibility of drug-pressured *P. falciparum* parasites.** (**a**) Selection process of W2 and Dd2 parasites subjected to AN3661 at either a single concentration or increasing concentrations over time. (**b**) Susceptibility to AN3661 of W2-strain wild-type (WT) and drug-pressured lines (replicates: WT, 13; W2-R1, 6; W2-R3, 3; W2-R3rev, 3; W2-R4, 6; W2-R4rev, 3; W2-R5, 2; W2-R5rev, 3). (**c**) Susceptibility to AN3661 of Dd2 strain WT and drug-pressured lines (replicates: WT, 27; Dd2-Ra, 9; Dd2-Rb, 11; Dd2-Rc, 3; Dd2-R1, 3; and Dd2-R2, 3). For all assays, details are in Supplementary Table 1, bar graphs represent mean ± s.e.m. IC$_{50}$ values, and *pfmdr1* copy numbers are indicated for each parasite line. Significance was determined using a two-tailed unpaired *t*-test, comparing drug-pressured parasites to parental strains, unless otherwise indicated. *$P < 0.05$, **$P < 0.005$, ***$P < 0.001$, ****$P < 0.0001$. n.s., not significant ($P > 0.05$); n.d., not determined.

homologue of subunit 3 of the CPSF complex, which has been well characterized in various eukaryotes[22]. In humans, this subunit has endonuclease activity and is referred to as CPSF-73, based on its molecular weight of 73 kDa. The predicted molecular weight of the *P. falciparum* homologue PfCPSF3 is 101 kDa. Dd2 parasites resistant to AN3661 had single PfCPSF3 mutations (T406I, Y408S, T409A or D470N; Fig. 4c). W2 parasites resistant to AN3661 demonstrated a more complex resistance trait: low-level resistant parasites had wild-type PfCPSF3, but high-level resistance was conferred by a single mutation, D470N, or a double mutation, H36Y/D470N.

We also applied whole-genome sequence analysis to revertant parasite lines that had been cultured without AN3661 after resistance selection. W2-R3rev, which showed wild-type sensitivity to AN3661 (mean IC$_{50}$ 23.4 nM), had lost the PfCPSF3 D470N mutation, but W2-R4rev, which remained resistant (IC$_{50}$ 3.0 μM), retained the double PfCPSF3 H36Y/D470N mutation. Low-level resistance to AN3661 observed in W2-R1 and W2-R2 was accompanied by amplification in the region of

chromosome 5 (position 946,695–971,095) that encompasses *pfmdr1* (PF3D7_0523000; Fig. 4b). This gene encodes an ABC transporter for which SNPs and CNV have been linked to altered sensitivity to a number of antimalarials[23].

**pfcpsf3 gene editing recapitulated AN3661 resistance.** To test the hypothesis that mutations in PfCPSF3 constitute the primary determinant of *P. falciparum* resistance to AN3661, we engineered Dd2 parasites to harbour either the T406I or Y408S mutations observed in our resistant lines. These experiments utilized a CRISPR-Cas9-based method of gene editing (Fig. 5a and Supplementary Table 7). Transfections yielded three parasite lines carrying the T406I mutation and two carrying the Y408S mutation. While most transfections included selection with 170 nM AN3661, importantly transfectant C4 (PfCPSF3 Y408S) did not, and rather was obtained using only WR99210 and blasticidin selection of the editing plasmids. All transfectants demonstrated 100% editing efficiency at the appropriate sites in

the genome, including silent mutations at the single-guide RNA (sgRNA)-directed Cas9 binding sites (Fig. 5b). In the engineered parasites, the PfCPSF3 T406I and Y408S mutations conferred a 12–50-fold decrease in AN3661 sensitivity, similar to changes in sensitivity in parasites that acquired these mutations after *in vitro* resistance selection (Fig. 5c). Selected and transfected parasites

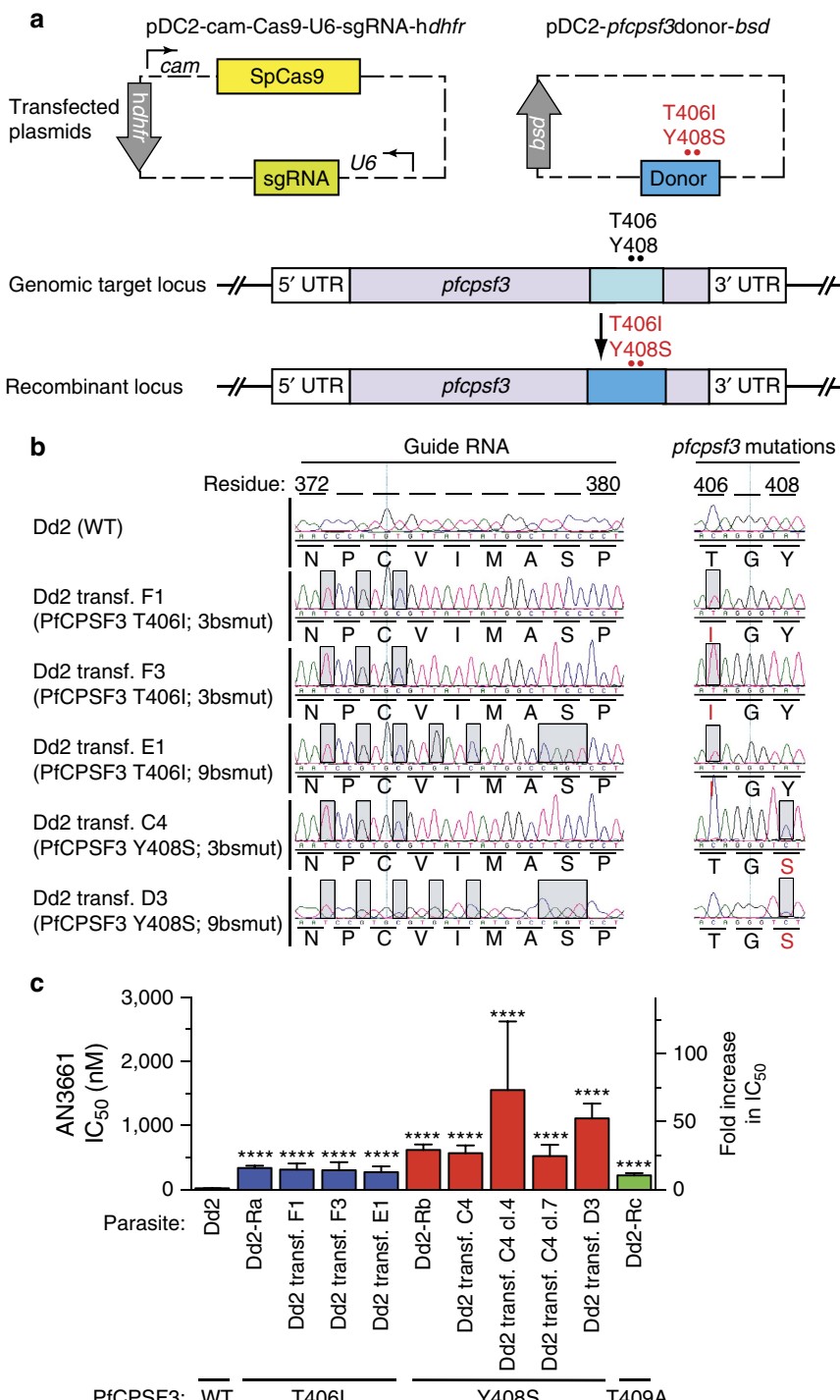

**Figure 5 | CRISPR-Cas9-mediated *pfcpsf3* editing. (a)** Dd2 parental (WT) parasites were transfected with a dual-plasmid strategy in which one plasmid encoded the sgRNA, Cas9 nuclease and a h*dhfr* selectable marker, and the other plasmid encoded a donor sequence containing either the T406I or Y408S PfCPSF3 mutations (in red), as well as synonymous mutations in the sgRNA-binding region. **(b)** Electropherograms showing unmodified Dd2 and genome-edited parasites. Grey boxes highlight nucleotides that differ from WT parasites. PfCPSF3 single-letter amino acid substitutions are indicated in red. **(c)** Susceptibility to AN3661 of the parental line (Dd2 WT), parasites selected *in vitro* (Dd2-Ra, Dd2-Rb and Dd2-Rc), and genetically modified lines (Dd2 transf. F1, F3, E1, C4, C4 cl.4, C4 cl.7 and D3; replicates: Dd2 WT, 27; Dd2-Ra, 9; Dd2 transf. F1, 4; Dd2 transf. F3, 4; Dd2 transf. E1, 4; Dd2-Rb, 11; Dd2 transf. C4, 5; Dd2 transf. C4 cl.4, 2; Dd2 transf. C4 cl.7, 2; Dd2 transf. D3, 9; Dd2-Rc, 3.). Assay details are in Supplementary Table 1. Bar graphs represent mean ± s.e.m. $IC_{50}$ values. The black bar denotes PfCPSF3 WT, blue bars T406I, red bars Y408S and green bar T409A. Significance was determined using a two-tailed unpaired *t*-test, comparing transfected parasites with the parental Dd2 strain. ****$P < 0.0001$.

with the Y408S mutation were consistently less sensitive to AN3661 compared to parasites with the T406I mutation. These results confirm a primary role for *pfcpsf3* mutations in conferring resistance to AN3661.

**Resistance mutations cluster around the enzyme active site**. Amino acid sequence alignment showed that PfCPSF3 is a close homologue of human CPSF-73 (Fig. 6a), with 61% similarity and 39% identity (calculations did not include gaps in the alignment due to plasmodial insertions). A crystal structure of human CPSF-73 (PDB code: 2I7T) revealed that the enzyme belongs to the zinc-dependent metallo-β-lactamase (MBL) family, with MBL and β-CASP domains, and an active site at the interface of these domains containing two zinc ions[24]. Since this crystal structure appears to be in an inactive conformation, with the β-CASP domain occluding the active site, we generated a new model based on a crystal structure of the messenger RNA (mRNA) processing ribonuclease TTHA0252 from *Thermus thermophiles* HB8 (PDB code: 3IEM), an enzyme with 47% similarity and 27% identity with PfCPSF3. This structure was in an active conformation complexed with an RNA analogue in the active site. In the *T. thermophiles*-based PfCPSF3 model the β-CASP domain does not occlude the bi-metal active site, which is accessible to substrates and inhibitors (Fig. 6b). In both models, the identified PfCPSF3 resistance mutations T406I, Y408S, T409A and D470N clustered around the active site (Fig. 6b). This finding suggests functional relevance of the resistance mutations and supports the genetic evidence implicating PfCPSF3 as the antimalarial target of AN3661.

**Molecular docking suggests oxaborole binding to PfCPSF3**. Considering the conformational complexity of the PfCPSF3 active site and limitations of forcefield parameters involving the boron atom, we docked AN3661 by matching important pharmacophore sites in the PfCPSF3 homology model based on *T. thermophiles*. The negatively charged tetrahedral oxaborole group from AN3661 was placed at the phosphate position at the cleavage site, interacting with the two catalytic zinc ions. Analogous interaction with metals has been seen in crystal structures of oxaboroles binding to PDE4 (ref. 16) and MBL NDM-1 (YR Freund, personal communication), with the oxaborole group acting as a transition state mimic and a phosphate mimic, respectively. The energy-minimized model of AN3661 at the active site of PfCPSF3 revealed the terminal carboxylate of AN3661 occupying an adjacent phosphate-binding site opposite R290 and Y252, forming a salt bridge and a hydrogen bond, respectively (Fig. 6b). Of the residues selected in AN3661-resistant parasites, Y408 and T409 were located at the entrance of the binding pocket in the β-CASP domain, in close proximity to the docked inhibitor, and D470 was positioned opposite Y408 in the RNA-metabolizing MBL domain (Fig. 6b). T406 was adjacent to R290, which directly interacts with the carboxylate from AN3661. It is interesting to note that in the *Toxoplasma gondii* CPSF-73 homologue, a Y328C (corresponding to Y252 in PfCPSF3) resistance mutation was selected under AN3661 pressure[25], further supporting the hypothesis that this residue is important in AN3361 binding.

**AN3661 inhibited the stability of *P. falciparum* transcripts**. The mammalian CPSF complex is required for the processing of newly synthesized transcripts (pre-mRNAs) to mature mRNA[24,26,27]. The cleavage of the 3′ end of pre-mRNA and subsequent addition of a poly(A) tail is necessary for mRNA stability and for export of mature mRNA from the nucleus to the cytosol, where it acts as a template for translation[26]. The observed

similarity between PfCPSF3 and human CPSF-73 led us to predict that inhibition of PfCPSF3 would impact the stability of parasite mRNA transcripts. As AN3661-treated parasites failed to progress through the trophozoite stage (Fig. 3a), we suspected that a defect in mRNA stability would be most apparent in trophozoites. We thus examined transcripts for three trophozoite-expressed genes, falcipain-2 (FP2), 1-cys peroxiredoxin and purine nucleoside phosphorylase (PNP)[28–30] in parasites cultured in the presence or absence of AN3661. Following a 4-h incubation with either AN3661 or the transcription initiation inhibitor actinomycin D, FP2, 1-cys peroxiredoxin and PNP transcripts were markedly reduced or undetectable in parental W2 and Dd2 parasites (Fig. 7a,e; Supplementary Figs 2 and 3). In contrast, parasites harbouring the PfCPSF3 mutations D470N, H36Y/D470N, T406I or Y408S selected with AN3661 or introduced with the CRISPR-Cas9 system demonstrated stable transcripts despite exposure to AN3661 (Fig. 7c,d,f–i). In all parasites examined, transcripts were unstable in the presence of actinomycin D, yet were unaffected by 4-h exposures to artemisinin or chloroquine, antimalarials that have unrelated modes of action (Fig. 7). Taken together, our data provide compelling evidence that AN3661 selectively inhibits PfCPSF3, leading to altered parasite mRNA processing and stability.

**Discussion**

Screening of a benzoxaborole library against cultured *P. falciparum* identified AN3661 as a potent antimalarial lead compound. Subsequent studies showed AN3661 to be active at low nanomolar concentrations against multiple *P. falciparum* strains that vary in their susceptibility to standard antimalarials and to be highly effective when administered orally to treat *P. berghei* and *P. falciparum* infections in mice. Multiple independently selected parasites with resistance to AN3661 contained mutations in the *P. falciparum* homologue of the endonuclease component of the CPSF complex that has been shown in other organisms to be responsible for pre-mRNA cleavage and polyadenylation[24]. Introduction of two of these PfCPSF3 mutations into wild-type parasites recapitulated resistance to AN3661, and modelling suggested that the mutations were in the PfCPSF3 active site at amino acids interacting with AN3661. Biochemical studies showed that treatment with AN3661 led to alterations in the stability of messages encoding three trophozoite proteins in wild-type, but not AN3661-resistant parasites. Collectively, these results support the conclusion that the primary target of AN3661 in *P. falciparum* is PfCPSF3.

CPSF is a multi-protein complex present in eukaryotic cells that is essential for processing of pre-mRNA to mRNA via cleavage and polyadenylation at the 3′-end of the pre-mRNA. In mammals, the coordinated activity of the CPSF complex and other proteins is initiated by interaction of CPSF-100 with an AAUAAA sequence located 10–30 bases upstream of the cleavage site[22,24,26,31,32]. Upon interaction with the AAUAAA sequence, the pre-mRNA undergoes site-specific endonucleolytic cleavage by CPSF-73 to generate the mRNA end, which then serves as a primer for synthesis of the poly(A) tail[26]. The endonuclease CPSF-73 is a member of the zinc-dependent MBL family, and contains 5 canonical MBL signature sequence motifs of histidine and aspartate residues, as well as a β-CASP domain[24,26,33]. The active site of CPSF-73 contains two zinc ions and is located at the interface of the MBL and β-CASP domains. PfCPSF3 is a homologue of mammalian CPSF-73 with conservation of the 5 MBL and β-CASP motifs.

Homology modelling of the structure of PfCPSF3 offered further support for its role as the target of AN3661. The

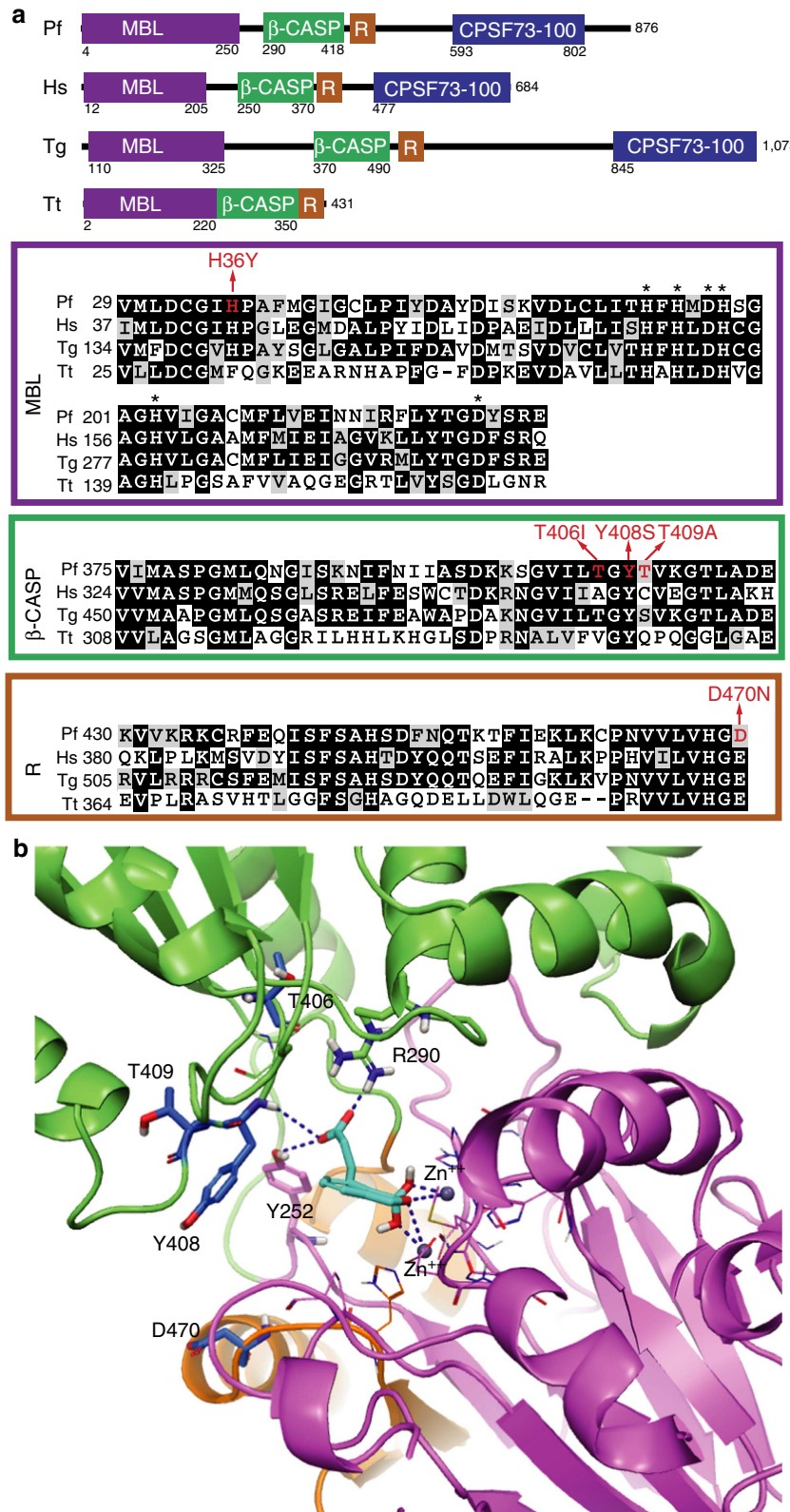

**Figure 6 | AN3661 resistance mutations reside in the MBL and β-CASP domains of PfCSPF3.** (**a**) Conserved domains predicted by NCBI for PfCPSF3 and its homologues in humans (Hs, gi:18044212), *Toxoplasma gondii* (Tg, KFG54681.1) and *T. thermophilus* (Tt, TTHA0252). R, RNA-metabolizing MBL domain. Amino acid numbers are shown for each protein. Asterisks denote amino acids predicted to bind the two zinc atoms in the MBL domain. Amino acids in red were mutated after drug selection. (**b**) Model of AN3661 (cyan) at the active site of PfCPSF3 (MBL domain, mauve; β-CASP domain, green, RNA-metabolizing MBL domain, orange), built based on the crystal structure of *T. thermophilus* TTHA0252 (PDB code: 3IEM). Residues D470, Y408, T409 and T406, which were mutated after selection by AN3661, are shown as blue stick models. The negatively charged benzoxaborole group interacts extensively with the two catalytic zinc ions (grey), and the carboxylate side chain interacts with R290 and Y252, forming a salt bridge and hydrogen bond, respectively.

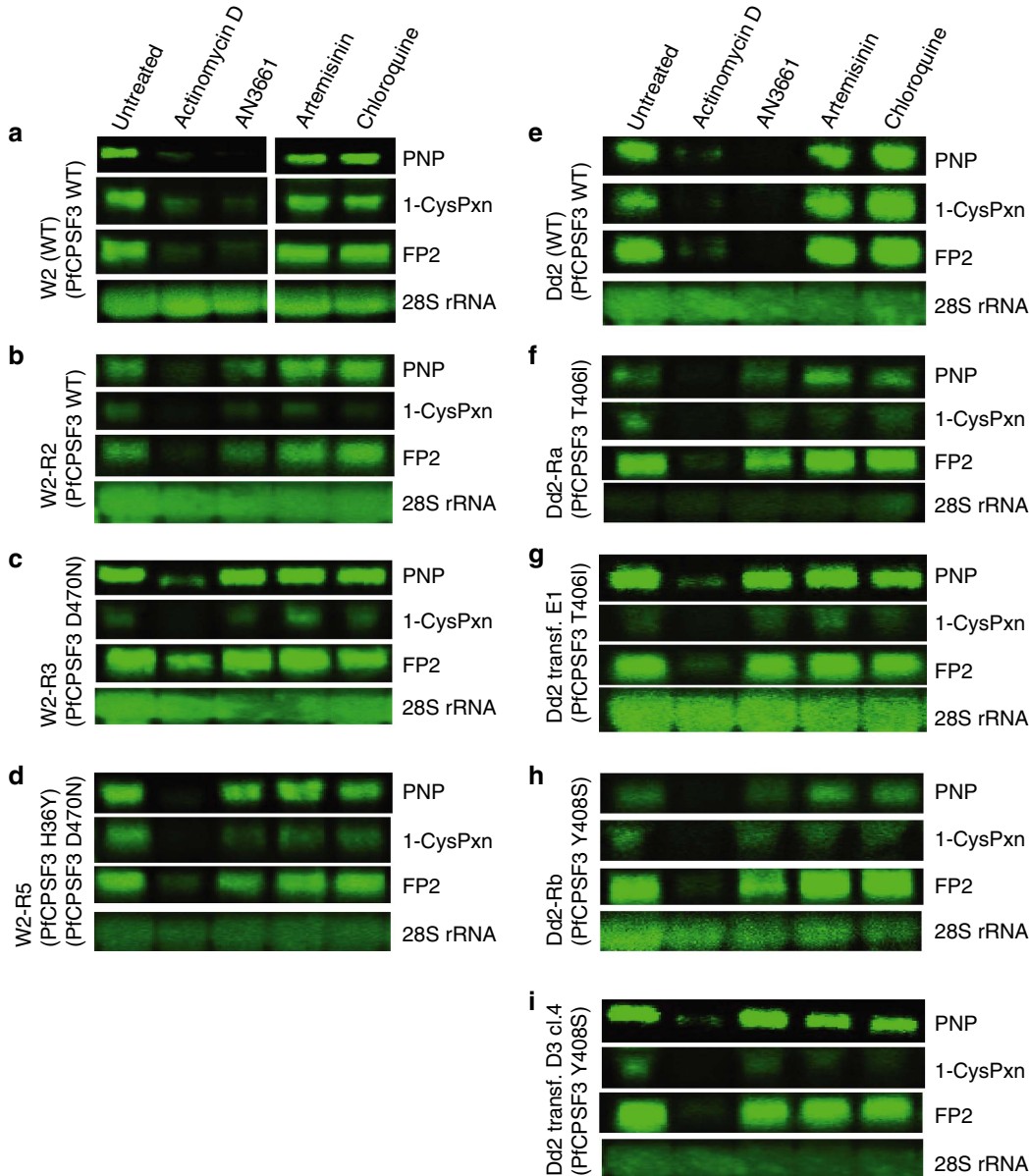

**Figure 7 | mRNA stability in the presence of AN3661.** Northern blots shown are of parasite lines labelled as defined in Figs 3 and 4 (**a–i**) that were treated for 4 h with DMSO (untreated), actinomycin D, AN3661, artemisinin or chloroquine. RNA was then processed and the transcripts for purine nucleoside phosphorylase (PNP), 1-cys peroxiredoxin (1-CysPxn) and FP2 were detected and imaged as described in 'Methods' section. Levels of 28S rRNA were examined as a loading control. The figure shows results from a single experiment. Two additional experiments yielded the same results.

mutations seen in *P. falciparum* after selection for decreased sensitivity to AN3661 were in residues predicted to be in the enzyme's active site and in proximity to AN3661. Remarkably, in parallel studies with *T. gondii*, a related apicomplexan parasite, selection of resistance to AN3661 was accompanied by mutations in the predicted active site of the *T. gondii* homologue of CPSF3 (ref. 25). Our docking model of AN3661 at the PfCPSF3 binding site is consistent with the structure-activity relationships established for this series of compounds as antimalarial agents. First, the oxaborole group is essential for activity, as replacement of a carbon atom for the boron resulted in an inactive compound[20]. Second, the carboxylic acid group is important for activity, although it can be replaced by other acidic groups such as a tetrazole[19]. The negatively charged tetrahedral oxaborole group serves as a unique ion chelator engaging the two zinc ions at the active site. The fact that AN3661 could be modelled into the bi-metal active site of PfCPSF3 is consistent with findings for

other benzoxaboroles, which have been demonstrated to bind to bi-metal centres. In the active site of PDE4, which contains a zinc and a magnesium ion, the boron atom of AN2898 binds to the activated water molecule in the bi-metal centre and mimics the transition state of the substrate cAMP[16]. In the co-crystal structure of NDM-1 MBL and a benzoxaborole β-lactamase inhibitor, AN4483, the boron atom is coordinated to the bi-zinc centre, similar to the predicted binding conformation for AN3661 in PfCPSF3 (YR Freund, personal communication). These examples demonstrate the potential of the oxaborole group in benzoxaboroles such as AN3661 to act as a metal chelator capable of binding to a variety of enzymes containing bi-metal centres.

In our parasite drug pressuring studies, low-level resistance to AN3661 was associated with increased copy number for *pfmdr1*, which encodes a presumptive ABC transporter[23]. Increased *pfmdr1* copy number has previously been associated with decreased sensitivity to the arylaminoalcohols mefloquine,

lumefantrine and halofantrine, and increased sensitivity to the 4-aminoquinolines chloroquine and amodiaquine[4]. Alterations in *pfmdr1* copy number may impact AN3661 accumulation at its intracellular site of action. However, high-level resistance to AN3661 was not associated with *pfmdr1* copy number, suggesting that alterations in PfCPSF3 are the dominant determinant for high-grade resistance.

Antimalarial drug discovery presents significant challenges. New antimalarials should meet multiple criteria, including excellent potency, ideally against multiple plasmodial species; oral bioavailability, ideally with cure after a single dose; safety in children and pregnant women; and low cost of production[34]. AN3661 shows progress in this direction as it displays outstanding potency against multiple strains of *P. falciparum in vitro* and is highly effective when administered orally in mouse models of *P. falciparum* and *P. berghei* infection. However, resistance was selected fairly readily *in vitro,* raising concern about clinically relevant selection of resistance. This concern may be obviated by incorporating PfCPSF3 inhibitors into combination regimens. The discovery of PfCPSF3 as the target for AN3361 enables us to rationally design inhibitors with improved drug-like properties and additional interactions that should reduce the propensity for resistance while maintaining selectivity.

## Methods

### Activity of compounds against cultured *P. falciparum*.
*P. falciparum* strains (3D7, W2, Dd2, K1, HB3, FCR3 and TM90C2B; obtained from the Malaria Research and Reference Reagent Resource Center) were cultured by standard methods at 2–3% haematocrit in Roswell Park Memorial Institute (RPMI)-1640 medium (Invitrogen) supplemented with 0.5% Albumax II (GIBCO), 2 mM L-glutamine, 100 mM hypoxanthine, 5 μg ml$^{-1}$ gentamicin, 28 mM NaHCO$_3$ and 25 mM HEPES at 37 °C in an atmosphere of 5% O$_2$, 5% CO$_2$ and 90% N$_2$, except that for parasite reduction ratio studies (Supplementary Fig. 1) parasites were cultured at 2% haematocrit in RPMI-1640 medium supplemented with 0.5% Albumax II, 2% D-sucrose, 0.3% glutamine and 150 mM hypoxanthine. *In vitro* dose-response assays were conducted in one of three ways. In the first, parasites were synchronized by treatment with 5% D-sorbitol and cultured in duplicate 96-well culture plates (200 μl per well) in the presence of serially diluted AN3661 or chloroquine (Sigma-Aldrich). Compound concentrations ranged from 0.056 to 1,000 nM. Control wells contained ≤0.2% dimethylsulphoxide (DMSO). After 48 h, the cultures were fixed with 2% formaldehyde for 24 h at 37 °C or 48 h at room temperature. Fixed cells were then stained with 4 nM YOYO-1 dye (Molecular Probes), and counts of treated and control cultures were determined using a fluorescence-activated cell sorter. In the second method, parasites that were mostly rings were seeded in 96-well culture plates at 1% haematocrit and 0.2% parasitemia. Parasites were subjected to a range of drug concentrations, with final DMSO concentrations ≤0.2%, for 72 h. Live parasites were stained with SYBR Green I (Invitrogen) and 100 nM MitoTracker Deep Red (Invitrogen), quantified by flow cytometry analysis on a BD Accuri C6 and analysed using FlowJo software. In the third method, parasites at 2% haematocrit and 0.5% parasitemia were exposed to a range of drugs for 48 h, and parasitemia was then quantified by [$^3$H]-hypoxanthine incorporation. IC$_{50}$ values were calculated by nonlinear regression using GraphPad Prism 6.0 software.

### Ex vivo dose response against *P. falciparum* field isolates.
The activity of AN3661 was tested against fresh *P. falciparum* isolates using an enzyme-linked immunosorbent assay directed against the histidine-rich protein-2, as previously described[35,36]. These isolates were collected in 2012 from children with *P. falciparum* malaria living in Tororo, Uganda, before receipt of drug treatment. The relevant clinical trials and analyses of cultured parasites were approved by the Uganda National Council of Science and Technology, the Makerere University Research and Ethics Committee, and the University of California, San Francisco Committee on Human Research.

### Cellular cytotoxicity assay.
Studied mammalian cells (all received from, authenticated by, and noted to be *Mycoplasma*-free by the American Type Culture Collection, see Supplementary Table 3) were seeded into 96-well plates at 2 × 10$^4$ cells per well in 100 μl growth medium (RPMI-1640 medium with 10% foetal bovine serum and 2 mM L-glutamine) with 10-fold serial dilutions (0.1 nM–100 μM) of AN3661; control wells had the same concentration of DMSO (0.25%). Plates were incubated at 37 °C in 5% CO$_2$ for 72 h. MTS (20 μl; 3-(4,5-dimethylthiazol-2-yl)-5-(3-carboxymethoxyphenyl)-2-(4-sulfophenyl)-2H-tetrazolium) was then added for 4 h, and absorbance was determined at 490 and

690 nm. Inhibition of cell viability was calculated based on the formula:
% inhibition of cell viability = $100 - (\Delta OD_{treated} / \Delta OD_{untreated}) \times 100$
($\Delta OD = OD_{490} - OD_{690}$).

### In vitro parasite reduction ratio.
This assay used limiting dilution to quantify the number of parasites that remained viable after various durations of treatment. *P. falciparum* 3D7 parasites (at 2% haematocrit and 0.5% parasitemia) were incubated with AN3661 for 24, 48, 72, 96 or 120 h. After appropriate time points, the compound was removed by washing, and parasites were serially diluted in 96-well plates and allowed to continue growing after drug wash-out. The number of viable parasites after various incubation times with compounds was determined after 21 and 28 days by measuring [$^3$H]-hypoxanthine incorporation in threefold serial dilutions of cultures, as previously reported[37]. Human biological samples were sourced ethically, and their research use was in accord with the terms of informed consents.

### Activity of AN3661 in murine models of malaria.
Protocols for studies of murine *P. berghei* infection were approved by the University of California, San Francisco Institutional Animal Care and Use Committee. Female Swiss Webster mice (6–8 weeks of age; 18–20 gm) were infected intraperitoneally with 6 × 10$^6$ *P. berghei*-infected erythrocytes (passaged from a donor mouse) and then treated, beginning 1 h after inoculation, with AN3661 (in 55% polyethylene glycol 300, 25% propylene glycol, 20% water) or chloroquine (in water) by oral gavage once daily for 4 days. Treatment group assignments were allocated randomly. Investigators were not blinded to treatment groups. Negative controls were treated with vehicle only. Infections were monitored by daily microscopic evaluation of Giemsa-stained blood smears. ED$_{90}$ values, based on comparisons of parasitemias between treated and control animals on the fourth day after the initiation of treatment, were calculated using GraphPad Prism 6.0 software. Mice were euthanized when parasitemias exceeded 50%.

Studies of murine *P. falciparum* infection were ethically reviewed and carried out in accordance with European Directive 2010/63/EU and the GSK Policy on the Care, Welfare and Treatment of Animals. *In vivo* efficacy against *P. falciparum* was conducted as previously described[21]. Age-matched female immunodeficient NOD.Cg-*Prkdc$^{scid}$ Il2rg$^{tm1Wjl}$*/SzJ mice (8–10 weeks of age; 22–24 gm) supplied by Charles River, L'Arbresle, France, under license of The Jackson Laboratory, Bar Harbor, ME, USA were engrafted with human erythrocytes (Red Cross Transfusion Blood Bank in Madrid, Spain) by daily intraperitoneal injection with 1 ml of a 50% haematocrit erythrocyte suspension (RPMI-1640 (Invitrogen), 25 mM HEPES (Sigma), 25% decomplemented AB$^+$ human serum (Sigma) and 3.1 mM hypoxanthine (Sigma)). The sample size per experimental group was 3, calculated as the minimum required to detect 50% reduction in parasitemia compared to a vehicle-treated control group, assuming a value for α (confidence level) of 0.05 and a value for β (power) of 0.9. Mice with ~40% circulating human erythrocytes were intravenously infected with 2 × 10$^7$ *P. falciparum* Pf3D7$^{0087/N9}$-infected erythrocytes (day 0). AN3661 was then administered by oral gavage for 4 consecutive days, beginning on day 3 after infection. AN3661 was prepared in 1% carboxymethylcellulose 0.1% Tween 80 in water before administration. Treatment group assignments were allocated randomly. Investigators were not blinded to treatment groups. Parasitemia was measured by flow cytometry in samples of peripheral blood stained with the fluorescent nucleic acid dye SYTO-16 (Molecular Probes; S-7578, 5 μM) and anti-murine erythrocyte TER119 monoclonal antibody (Becton Dickinson 553673; 10 μg ml$^{-1}$) in serial 2 μl blood samples taken every 24 h until assay completion. The ED$_{90}$ was estimated by fitting a four parameter logistic equation using GraphPad Prism 6.0.

### Stage specificity assay.
The stage-specific activity of AN3661 and chloroquine was analysed as previously described[10,38]. Highly synchronous W2 strain *P. falciparum* (synchronized by treatment with 5% D-sorbitol) were cultured in triplicate wells in 96-well culture plates with 370 nM AN3661 and 1.3 μM chloroquine for 8 h intervals, beginning at the ring stage. Control cultures contained equivalent concentrations of DMSO (<0.2%). At the end of each interval, the cultures were washed three times and resuspended in culture media without drug. After 48 h, when control parasites were at the ring stage, the cultures were fixed with 2% formaldehyde and counted using flow cytometry as detailed above, and parasitemias were compared with those of untreated control parasites.

### Selection of parasites with decreased sensitivity to AN3661.
For step-wise selection, triplicate 10 ml cultures containing 6 × 10$^7$ asynchronous W2 strain or a clonal population of the Dd2 strain were incubated with step-wise increasing concentrations of AN3661. W2 parasites were subjected to 37 nM AN3661 for 43 days, 74 nM for 43 days, 300 nM for 90 days, 1 μM for 73 days and finally 5 μM for 73 days. Dd2 parasites were subjected to 37 nM AN3661 for 31 days, then 200 nM for 73 days. For single-step resistance selection with Dd2, 2 × 10$^9$ parasites in duplicate or 2 × 10$^7$ parasites were incubated with 170 nM AN3661. For the first six days, parasitemia was monitored by microscopy and media containing drug was changed daily. Then, media was changed every 2 days, and fresh erythrocytes were replaced once a week. For all selections, parasitemia was monitored by microscopy, and recrudescent parasites were cloned by limiting dilution.

**Whole-genome sequencing.** To prepare genomic DNA, synchronized *P. falciparum*-infected erythrocytes (100 ml, 2% haematocrit and 10% parasitemia) were treated with 0.15% saponin for 5 min on ice to lyse erythrocytes, followed by 3 washes in PBS. Parasite pellets were lysed in 150 mM NaCl, 10 mM EDTA, 50 mM Tris-HCl pH 7.5, 0.1% sarkosyl (Sigma-Aldrich) and 200 mg ml$^{-1}$ proteinase K (Qiagen) overnight at 37 °C. The samples were then subjected to extraction with phenol/chloroform/isoamyl alcohol (25:24:1) pH 7.9 (Ambion), treatment with 0.05 mg ml$^{-1}$ RNAse A (1 h at 37 °C), two additional phenol/chloroform extractions, one chloroform extraction and then ethanol precipitation. All phenol/chloroform extractions were done using light phase lock tubes (5 Prime). Genomic DNA libraries were prepared from 100 ng DNA using the Nextera DNA Sample Prep Kit (Illumina) according to the manufacturer's instructions, with the exception that the number of cycles was 6 and the bridge amplification step was at 65 °C for 6 min[39]. Each library was barcoded with unique sets of two indices from the Nextera Index Kit (Illumina) to allow multiple samples to be run on one flow cell. Next, fragments of 360–560 bp were extracted and collected using Lab Chip XT (Caliper Life Sciences) according to the manufacturer's instructions. The fragments were amplified by limited-cycle PCR using Kapa HiFi DNA polymerase (Kapa Biosystems) with dNTPs with an 80% AT coding bias (6 cycles of 95 °C for 10 s, 58 °C for 30 s, 65 °C for 6 min). Primers used for both PCR steps are detailed in Supplementary Table 8. Library concentrations were determined with a DNA Bioanalyzer (Agilent), and libraries were pooled at concentrations of 2 nM per library. Library preparations were then completed[39] and sequenced at the UCSF Center for Advanced Technology, followed by sequencing on a HiSeq 2000 system (Illumina). Sequence data for each library were aligned to the 3D7 reference genome using Bowtie[40], discarding reads with >1 nucleotide mismatch or multiple alignments across the genome. For the identification of SNPs, reads were matched to those from the parental strain, and the top 200 SNPs per chromosome, ranked according to frequency of conflicting nucleotides per position in the genome, were chosen and filtered based on standard parameters[39]. SNPs were characterized as legitimate if the number of reads covering the position was >10 and the frequency of the SNP was at least 80%. Searches for novel SNPs included only non-synonymous SNPs in exons, excluding hypervariable genes (*var*, *rifin* and *stevor*). CNV was analysed using the UCSC Genome Browser[41].

**Dideoxy sequencing.** Genomic DNA was extracted using the QIAamp DNA mini kit (Qiagen) according to the manufacturer's instructions. The *pfcpsf3* gene (listed as PF3D7_1438500 on PlasmoDB) was amplified in 3 fragments using the Phusion Hot Start II High-Fidelity DNA Polymerase kit (Thermo Scientific) with 80% AT dNTPs and primers (Supplementary Table 8; 95 °C for 3 min, 30 cycles of 95 °C for 10 s, 52 °C for 30 s, 65 °C for 1 min and a final extension on 65 °C for 10 min). The amplified fragments were cleaned using ExoSAP-IT (Affymetrix), mixed with sequencing primers, and sequenced at the UCSF Genome Core Facility.

**Determination of *pfmdr1* copy number by quantitative PCR.** Multiplex PCR was performed in MicroAmp 96-well plates (Applied Biosystems) in 25 μl reactions containing TaqMan mastermix buffer (8% glycerol, 0.625 U DNA polymerase, 5.5 mM MgCl$_2$, 300 μM dNTP, 600 nM), reaction reference dye ROX (5-carboxy-X-rhodamine), pH 8.3, 300 nM of each forward and reverse primer, 100 nM of each probe and 2.5 μl DNA template. The amplification protocol was 95 °C for 10 min, followed by 50 cycles of 95 °C for 15 s and 58 °C for 1 min using an Applied Biosystems 7500 Real Time PCR machine. β-tubulin was used as the internal reference. Fluorescence data were analysed by a comparative C$_t$ method comparing changes in fluorescence signal of the target (*pfmdr1*) relative to the internal reference (β-tubulin)[42]. All signals were normalized to the passive reference signal (ROX). The detection threshold was set above the mean baseline value for fluorescence of the first 15 cycles. Primer sequences are detailed in Supplementary Table 8.

**CRISPR-Cas9-mediated editing of *pfcpsf3* mutations.** Choice of sgRNA, plasmid construction and parasite transfections were performed as described in ref. 43. Briefly, the sgRNA 5′-GGGAAGCCATAATAACACATGGG-3′ (PAM site underlined) was identified with the help of the Broad Institute's sgRNA Designer (https://www.broadinstitute.org/rnai/public/analysis-tools/sgrna-design). The chosen sgRNA was close to the mutations of interest (78 and 82 bp away), had a high sgRNA score (0.68), and was unique in the genome (https://chopchop.rc.fas.harvard.edu). The sgRNA construct with overhanging BbsI sites was inserted into a pDC2-cam-Cas9-U6-h*dhfr* plasmid[43] that encodes *Streptococcus pyogenes* Cas9 expressed under the calmodulin promoter and a human *dhfr* selectable marker (that mediates resistance to WR99210); the sgRNA was expressed from a U6 promoter. A ∼1.4 kb fragment of *pfcpsf3* was amplified from Dd2 (*pfcpsf3* wt), Dd2-Ra (*pfcpsf3* C1217T) and Dd2-Rb (*pfcpsf3* A1223C) parasites using primers p5150 and p5151, detailed in Supplementary Table 8. Using the Q5 Site-Directed Mutagenesis Kit (NEB), three or nine nucleotide substitutions were introduced into the donor sequences at sgRNA-binding sites to protect from further Cas9 recognition and cleavage (binding site mutations, detailed in Supplementary Table 7). The resulting fragment was inserted using ApaI and BamHI restriction sites into the pDC2-*bsd* plasmid[43], which expresses the blasticidin S-deaminase (*bsd*) selectable marker. Each of the pDC2-*pfcpsf3* donor

-*bsd*-CPSF3donor plasmids encoding PfCPSF3 WT, T406I or Y408S was transfected eight times into Dd2 parasites by electroporation, along with the pDC2-cam-Cas9-U6-sgRNA-h*dhfr* plasmid. Of those eight transfections per plasmid, two were maintained in 2.5 nM WR99210 and 2 μg ml$^{-1}$ blasticidin, another two were exposed to 2.5 nM WR99210 and 2 μg ml$^{-1}$ blasticidin for 6 days and then maintained in drug-free media, and four were maintained in 170 nM AN3661. In total, five lines incorporated the PfCPSF3 mutations, along with silent binding site mutations. Of these 5 lines, one was the result of selection for the plasmids using 2.5 nM WR99210 and 2 μg ml$^{-1}$ blasticidin, and the other four were selected with 170 nM AN3661, as detailed in Supplementary Table 7.

**Homology modelling of PfCPSF3 and docking analysis of AN3661.** Sequences were from NCBI for human CPSF-73 (gi:18044212), PlasmoDB for PfCPSF3 (PF3D71438400) and PDB for *T. thermophiles* TTHA0252 HB8 (PDB code: 3IEM). Sequences were aligned using the NCBI BLAST Clustal Omega method and Maestro Prime (Schrodinger, LLC). Similarity and identify scores were calculated using Maestro, not including alignment gaps due to PfCPSF3 insertions. Crystal structures of human CPSF-73 (PDB code: 2I7T) and *T. thermophiles* HB8 (PDB code: 3IEM) served as templates to build homology models of PfCPSF3 using Maestro. The tetrahedral form of AN3661 was generated using Optimized Potentials for Liquid Simulations 2005 forcefield[44] and manually docked into the active site of the TTHA0252-derived homology model of PfCPSF3, mimicking the binding mode of benzoxaborole analogues observed from the crystal structures of MBL NDM-1 (YR Freund, personal communication). The final docking model was obtained by minimizing the complex using a water solvation model and maintaining a fixed protein backbone.

**RNA stability assay.** Ring stage *P. falciparum* W2 (10–12 h post-invasion) or Dd2 (16–18 h post-invasion) parasites were cultured at 10–15% parasitemia and 2% haematocrit in a volume of 20 ml. Cultures were exposed to 400 nM AN3661, 600 nM chloroquine, 70 nM artemisinin or actinomycin D at either 80 μg ml$^{-1}$ for W2 or 320 μg ml$^{-1}$ for Dd2. Control cultures were exposed to DMSO at <0.01% final concentration. After 4 h of drug treatment, parasites were frozen in liquid nitrogen and stored at −80 °C. Total RNA was subsequently extracted using the Trizol Plus RNA purification kit (Purelink RNA mini kit, Ambion) according to the manufacturer's instructions. Briefly, the frozen parasitized RBC pellets were dissolved in pre-warmed Trizol reagent, followed by chloroform extraction. The aqueous phase was mixed with ethanol to obtain a final ethanol concentration of 35% and total RNA was bound, washed, and eluted from the provided spin cartridges according to the manufacturer's instructions. For Northern blots, 20 μg of total RNA was separated on 1.3% formaldehyde agarose gel and transferred to a Hybond N+ membrane (Amersham) overnight through capillary action. For hybridization, RNA was detected using DNA probes labelled with biotin-16-dUTP (Roche Diagnostics) (Supplementary Table 8). Northern blot signal was detected using Typhoon Trio imager (Amersham biosciences) and quantified using ImageQuant software.

**Data availability.** All relevant data are available from the corresponding author upon request.

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

## Acknowledgements

This work was supported by the National Institutes of Health (AI095324 and AI103058) and the Medicines for Malaria Venture. J.D. was funded by the Howard Hughes Medical Institute.

## Author contributions

All authors conceived and designed the experiments; E.S., C.L.N., M.C.S.L., D.G., L.M.S., J.G., J.L. and R.A.C. acquired the data; E.S., C.L.N., Y.Z., V.A., L.M.S., R.A.C., F.J.G., J.D., Y.R.F., D.A.F. and P.J.R. analysed and interpreted the data; E.S., C.L.N., Y.Z., M.R.K.A., Y.R.F., D.A.F. and P.J.R. played primary roles in writing the manuscript. All authors approved the final manuscript.

## Additional information

**Competing financial interests:** Y.-K.Z., Y.Z., M.R.K.A., C.D. and Y.R.F. are employees of Anacor Pharmaceuticals, Inc. L.M.S., M.J.L.M. and F.J.G. are employees of GlaxoSmithKline. The remaining authors declare no competing financial interests.

