## [Peer Review File · Nature Communications]

Reviewers' comments:

Reviewer #1 (Remarks to the Author):

The two papers (NCOMMS-16-21360 and NCOMMS-16-21336) present CPSF3 as a novel drug target for plasmodium falciparum and Toxoplasma gondii. Both of these diseases demand new and effective therapies to tackle the shortcomings of the current treatments – as such the work is important and of great interest to everyone from basic scientists through to clinicians. The use of boron containing compounds for therapeutic benefit is a growing area already with notable successes. The nature of these compounds are potentially opening up new drug targets across diseases which have been challenging to date.

One compound (AN3661) has been revealed as potent against both indications: As an antimalarial very good levels of activity against lab strains and also field isolates together with impressive performance in a Plasmodium berghei and Plasmodium falciparum mouse models. As a toxoplasmodial potent in vitro activity is reported.

A large amount of biological work provides very strong evidence that CPSF3 is the target for the benzoxaborole AN3661 for both indications.

All of the work is very clearly presented and described and is concise. The interpretations and conclusion from data presented are entirely appropriate. The references used looked apt.

The impact in the toxoplasmosis field may well be larger than the malarial field, due to the relative large number of projects under development, such as those those supported by MMV. (Other reviewers more au fait with the toxoplasmosis field may have comments in this respect).

I have been asked specifically to comment on the molecular modelling. All approaches described and used seemed completely suitable. However, inclusion of a statement in the methods that it was the tetrahedral version of AN3661 which was docked (with the ligated OH taking up the fourth position around the boron group) should be made explicit. This will also need to be reflected in the malaria paper on page 8. In the toxoplasmosis paper figure 3c may not be correctly rendered – I could not make out the fourth group (OH?) around the boron?

Subject to these minor issues being satisfactorily addressed I recommend both paper should be published.

Reviewer #2 (Remarks to the Author):

The manuscript by Sonoiki et al describes efforts to identify the target of a benzoxaborole compound that has good antimalarial activity. The same authors have published a number of papers on the benzoxaborole series and have identified several potent compounds with

good in vivo activity. The paper is important because it identifies the target (cleavage and polyadenylation specificity factor (CPSF)) of at least one of these compounds (AN3661) but also because the work has identified a new target for malaria drug discovery that wasn't previously known. Thus at least AN3661 hits a new target that doesn't share cross resistance with known antimalarials. The target ID work has been carefully done, first by identifying the likely target by selecting for resistant strains and then performing whole genome sequencing, but then by careful validation of the mutant phenotypes using CRISPR technology. Biochemical work was also performed to provide further demonstration of the target by showing decreased levels of mRNA biosynthesis. I do have a number of minor comments that should be address prior to publication.

- 1) The same authors have published several papers on benzoxaboroles with antimalarial activity, including a paper describing the in vitro antimalarial activity of AN3661. These papers need to be referenced in the introduction and not just in the discussion. Line 61/62 in the introduction states the compound was identified in a screen of a benzoxaborole library without any reference. The other published papers by the authors on this topic should be referenced at this point.
- 2) The same authors have published on compounds from the series that have greater in vitro potency than AN3661 (J Med Chem. 2015 Jul 9; 58(13): 5344–5354.), it would be useful to know if these compounds show the same mechanism of action as AN3661.
- 3) Line 72, it's not really correct to say no apparent cytotoxicity when an EC50 of 60 μ M was measured on a mammalian cell line and when the data on other cell lines was collected to variable top concentrations (some as low as 25 μ M). So it would be better to just report the selectivity range.
- 4) Line 89. They report an AUC at the ED90 in the SCID study, but an AUC over what time period? Also there is no methods section for the PK part of the study.
- 5) The section describing the docked structure of AN3661 to the target could be shorter. The model may be useful but it is only a model based on a structure of a homologous protein so many of the fine details are likely to prove to be different once the actual structure in complex to the inhibitor was determined.
- 6) The accession codes for the CPSF homolog used for the modeling need to be stated in the figure 5 legend.
- 7) Line 464 seems to suggest that an unpublished structure was used to develop the modeled structure, which is problematic since that means the methods can't be fully described.
- 8) An ethics statement for the animal work should be added in the Methods section.
- 9) The statistical analysis of the data for most figures and tables is not full explained and may be incorrect. None of the tables or figures indicate the number of replicates that went into determining the various EC50 data, nor is the meaning of the error bars in graphs defined. The only statement on page 13 line 307 says duplicate data were collected. Standard deviations and standard error of the mean can't be calculated from duplicate data. So number of replicates need to be stated clearly in each table and figure legend and if less than 3, than the error bars should be changed to represent range (not SD or SEM) and data in tables (e.g. supplemental tables S1 and S2) should report the values of the two replicates for those cases.
- 10) Too many significant figures are reported in some of the tables, and variable numbers

are reported ranging anywhere from 2 to 5. Two significant figures should be used consistently for IC50 data.

Reviewer #3 (Remarks to the Author):

Per the editor's request, the two studies by Sonoiki, et al. and by Palencia, et al. have been considered jointly. Detailed comments relevant to each study are listed below. The two studies detail the effectiveness of a benzoxaborole AN3661 against *Toxoplasma* and *Plasmodium*. A major aim of both manuscripts is to argue for a mechanism of action that involves inhibition of the cleavage and polyadenylation specificity factor subunit 3 (CPSF3) in these parasites. In the absence of direct enzymatic assays—beyond the scope of this work—the best evidence for the proposed mechanism of action is (1.) recovery of AN3661-resistant parasites bearing mutations in the CPSF3 gene, and, in the case of the Sonoiki study, (2.) demonstration of mRNA loss in response to AN3661 for WT but not resistant strains of *P. falciparum*. The effectiveness of the first line of evidence is undercut by the manner of its confirmation: testing whether specific mutations in CPSF3 cause resistance requires that the mutations be introduced WITHOUT AN3661 selection, and once the appropriate rearrangement has been confirmed, testing for resistance. All the mutants in the Palencia paper, and most of the mutants in the Sonoiki paper were isolated by selecting the transfected parasite populations with AN3661, making it tautological to later observe resistance in the resulting clones, and confounding their interpretation. In other words, the only proper experimental confirmation that mutations in CPSF3 can cause AN3661 resistance is provided by Sonoiki et al. through the generation and analysis of the transfectant population C4. I therefore strongly suggest that the two studies be merged in order to proceed with publication at Nature Communications, taking into account the following reasons: (1.) identification of CPSF3 mutations by Palencia et al. rely on the prior identification of such mutations by Sonoiki et al., (2.) demonstration that the CPSF3 mutations were causal was not appropriately performed by Palencia et al., (3.) Palencia et al. provide important evidence for the function of the compound in an in vivo model of toxoplasmosis, and (4.) Sonoiki et al. provide the only analysis of the molecular role of CPSF3 on mRNA levels by performing the Northern blots presented in figure 6. The pharmacological properties of this compound, its activity at low nM concentrations, and its confirmed utility in mouse models of toxoplasmosis, highlight the importance of this work and I hope that the authors will find a way to merge the studies.

Specific Comments:

1. The general presentation of the figures could be improved by merging Fig. 1 and Fig. 2, and including the *P. berghei* data in the main figure. I think the latter is important because it provides an in vivo validation of the compound's effect.
2. Exploring the correlation between the different mutations in Fig. 3 and Fig. 4 might strengthen the case for their effect, since it appears that the both spontaneous and targeted mutations of Y408 have stronger effects than mutation of other CPSF3 residues.

3. Fig. 6 would be significantly strengthened by providing quantification of the Northern blots across several experimental replicates for WT and C4 parasites.

Thank you for your review of MS# NCOMMS-16-21360, "A potent antimalarial benzoxaborole targets a *Plasmodium falciparum* cleavage and polyadenylation specificity factor homologue." This response refers only to this manuscript, and not the related *Toxoplasma* manuscript (NCOMMS-16-21336) that was submitted together with our manuscript. We respond to review comments in a point-by-point manner below. Page numbers below refer to the untracked version of the manuscript. We also made some changes to correct minor errors and to meet all requirements for the *Nature Communications* checklist (e.g. some subheadings were shortened to meet the 60 character limit and some details were added for animal model and cell line experiments); these are all highlighted on the Track Changes version of the revised manuscript.

Reviewer #1:

1) The two papers (NCOMMS-16-21360 and NCOMMS-16-21336) present CPSF3 as a novel drug target for *Plasmodium falciparum* and *Toxoplasma gondii*. Both of these diseases demand new and effective therapies to tackle the shortcomings of the current treatments – as such the work is important and of great interest to everyone from basic scientists through to clinicians. The use of boron containing compounds for therapeutic benefit is a growing area already with notable successes. The nature of these compounds are potentially opening up new drug targets across diseases which have been challenging to date. One compound (AN3661) has been revealed as potent against both indications: As an antimalarial very good levels of activity against lab strains and also field isolates together with impressive performance in a *Plasmodium berghei* and *Plasmodium falciparum* mouse models. As a toxoplasmodial potent in vitro activity is reported. A large amount of biological work provides very strong evidence that CPSF3 is the target for the benzoxaborole AN3661 for both indications. All of the work is very clearly presented and described and is concise. The interpretations and conclusion from data presented are entirely appropriate. The references used looked apt.

Reply: We thank the reviewer for these supportive comments.

2) I have been asked specifically to comment on the molecular modelling. All approaches described and used seemed completely suitable. However, inclusion of a statement in the methods that it was the tetrahedral version of AN3661 which was docked (with the ligated OH taking up the fourth position around the boron group) should be made explicit. This will also need to be reflected in the malaria paper on page 8.

Reply: This comment refers principally to the *Toxoplasma* manuscript. We note that the original version of our manuscript mentioned the tetrahedral form of AN3661 twice in Results ("The negatively-charged tetrahedral oxaborole group from AN3661 was placed at the phosphate position at the cleavage site, interacting with the two catalytic zinc ions.", P. 9; "The negatively charged tetrahedral oxaborole group serves as a unique ion chelator...", P. 12) and in Methods ("The tetrahedral form of AN3661 was generated..."; P. 21). These sentences have been retained.

3) Subject to these minor issues being satisfactorily addressed I recommend both paper should be published.

Reply: We thank the reviewer for these supportive comments.

Reviewer #2:

1) The manuscript by Sonoiki et al describes efforts to identify the target of a benzoxaborole compound that has good antimalarial activity. The same authors have published a number of papers on the benzoxaborole series and have identified several potent compounds with good in

vivo activity. The paper is important because it identifies the target (cleavage and polyadenylation specificity factor (CPSF)) of at least one of these compounds (AN3661) but also because the work has identified a new target for malaria drug discovery that wasn't previously known. Thus at least AN3661 hits a new target that doesn't share cross resistance with known antimalarials. The target ID work has been carefully done, first by identifying the likely target by selecting for resistant strains and then performing whole genome sequencing, but then by careful validation of the mutant phenotypes using CRISPR technology. Biochemical work was also performed to provide further demonstration of the target by showing decreased levels of mRNA biosynthesis. I do have a number of minor comments that should be address prior to publication.

Reply: We thank the reviewer for these supportive comments.

2) The same authors have published several papers on benzoxaboroles with antimalarial activity, including a paper describing the in vitro antimalarial activity of AN3661. These papers need to be referenced in the introduction and not just in the discussion. Line 61/62 in the introduction states the compound was identified in a screen of a benzoxaborole library without any reference. The other published papers by the authors on this topic should be referenced at this point.

Reply: As requested we have added three references (all of our papers on antimalarial activity of benzoxaboroles) to the Introduction. Reference 12 was not previously cited. References 19 and 20 were cited in our Discussion, but are now cited where we mention the screen of the benzoxaborole library at the end of the Introduction, and again at the beginning of Results, where we describe the results of our screen.

3) The same authors have published on compounds from the series that have greater in vitro potency than AN3661 (J Med Chem. 2015 Jul 9; 58(13): 5344–5354.), it would be useful to know if these compounds show the same mechanism of action as AN3661.

Reply: This is a very appropriate question, but sorting out mechanisms of multiple classes of benzoxaboroles is an ongoing project that is not ready for definitive description. In brief, we previously published on benzoxaboroles that target leucyl tRNA synthetase (reference 10), we describe here action of AN3661 against CPSF3, and our current candidate, AN13762, and some other benzoxaboroles under study clearly have different targets, but the specific targets have not yet been identified. We believe that it is best not to speculate further about targets of other benzoxaboroles in this manuscript. Subsequent publications will address the remarkable ability of different benzoxaboroles to target a broad range of *P. falciparum* targets.

4) Line 72, it's not really correct to say no apparent cytotoxicity when an EC₅₀ of 60 uM was measured on a mammalian cell line and when the data on other cell lines was collected to variable top concentrations (some as low as 25 uM). So it would be better to just report the selectivity range.

Reply: We have changed the relevant sentence (P. 5) to the following: "AN3661 showed minimal cytotoxicity against mammalian cell lines, with a CC₅₀ 60.5 μM against Jurkat cells, and all other CC₅₀ values greater than the highest concentrations tested (25 μM or above)."

5) Line 89. They report an AUC at the ED₉₀ in the SCID study, but an AUC over what time period? Also there is no methods section for the PK part of the study.

Reply: We regret that this sentence was included in error. We originally included a larger section on AN3661 PK, but then elected to omit all description of PK results, as we determined that these were too preliminary for publication, and not directly relevant to our main thesis. The sentence noticed by our reviewer was retained by accident. We have now omitted this sentence, and so results and methods for PK studies are no longer relevant.

6) The section describing the docked structure of AN3661 to the target could be shorter. The model may be useful but it is only a model based on a structure of a homologous protein so many of the fine details are likely to prove to be different once the actual structure in complex to the inhibitor was determined.

Reply: We appreciate the reviewer's concerns, but we worked hard to write a concise, but adequately detailed description, and we do not think that it will be improved by condensing it. The fact that this is "only a model" is clearly described in our manuscript.

7) The accession codes for the CPSF homolog used for the modeling need to be stated in the figure 5 legend.

Reply: To satisfy this appropriate request we changed the title of Fig. 5b (now Fig. 6b) to "Model of AN3661 (cyan) at the active site of PfCPSF3 (MBL domain, mauve; β -CASP domain, green, RNA-metabolizing MBL domain, orange), built based on the crystal structure of *Thermus thermophilus* TTHA0252 (PDB code: 3IEM)."

8) Line 464 seems to suggest that an unpublished structure was used to develop the modeled structure, which is problematic since that means the methods can't be fully described.

Reply: To clarify, our binding mode assumption was based on the published structure of PDE4 (reference 16); the unpublished structure of NDM-1 served to support this assumption. This structure will be published separately. To help readers appreciate this subtle distinction, we edited the manuscript to mention results for the published structure before that of the unpublished structure.

9) An ethics statement for the animal work should be added in the Methods section.

Reply: We apologize for this oversight. We have added the following two sentences to support in vivo studies with *P. berghei* and *P. falciparum* (section of Methods entitled "Activity of AN3661 in murine models of malaria", P. 15-16). "Protocols for studies of murine *P. berghei* infection were approved by the University of California, San Francisco Institutional Animal Care and Use Committee." "Studies of murine *P. falciparum* infection were ethically reviewed and carried out in accordance with European Directive 2010/63/EU and the GSK Policy on the Care, Welfare and Treatment of Animals." Of note, some additional information regarding animal model studies was added based on guidance from the reporting checklist.

10) The statistical analysis of the data for most figures and tables is not fully explained and may be incorrect. None of the tables or figures indicate the number of replicates that went into determining the various EC50 data, nor is the meaning of the error bars in graphs defined. The only statement on page 13 line 307 says duplicate data were collected. Standard deviations and standard error of the mean can't be calculated from duplicate data. So number of replicates need to be stated clearly in each table and figure legend and if less than 3, then the error bars should be changed to represent range (not SD or SEM) and data in tables (e.g. supplemental tables S1 and S2) should report the values of the two replicates for those cases.

Reply: We regret our oversights. The requested information was already provided in many figures and tables, but clarification was needed. For Figs. 3 and 4 (now Figs. 4 and 5) we neglected to note the following, now added to each legend: "Assay details are in Supplementary Table 1."; these details include number of assays and replicates. Information on the number of replicate experiments and meaning of the error bars was added as needed to Supplementary Figs. 1, 2, and 3. Responding to the concern about number of replicates for statistical analysis, we quote the web site for GraphPad Prism, which was used for statistical analyses. "Is it valid to calculate the SD or SEM or CI of two values? It seems to be common lab folklore that the calculations of SD or SEM are not valid for $n=2$. This folklore is wrong. The equations that

calculate the SD, SEM and CI all work just fine when you have only duplicate (N=2) data.” A long discussion on this web site entitled “Simulations to prove that the SD and SEM calculations work for n=2” includes acknowledgement that uncertainty is high when n=2, but that for 10,000 data set simulations, the average of computed variances was within 1% of the true population variance. Thus, “the variance computed from n=2 data is a valid assessment of the scatter in your data, no less valid than a variance computed from data with larger n.”

11) Too many significant figures are reported in some of the tables, and variable numbers are reported ranging anywhere from 2 to 5. Two significant figures should be used consistently for IC50 data.

Reply: We agree. For tables 1 and 2, we have corrected inconsistent use of significant figures. We now include a maximum of 3 significant figures.

Reviewer #3:

1) Per the editor’s request, the two studies by Sonoiki, et al. and by Palencia, et al. have been considered jointly. Detailed comments relevant to each study are listed below. The two studies detail the effectiveness of a benzoxaborole AN3661 against *Toxoplasma* and *Plasmodium*. A major aim of both manuscripts is to argue for a mechanisms of action that involves inhibition of the cleavage and polyadenylation specificity factor subunit 3 (CPSF3) in these parasites. In the absence of direct enzymatic assays—beyond the scope of this work—the best evidence for the proposed mechanism of action is (1.) recovery of AN3661-resistant parasites bearing mutations in the CPSF3 gene, and, in the case of the Sonoiki study, (2.) demonstration of mRNA loss in response to AN3661 for WT but not resistant strains of *P. falciparum*. The effectiveness of the first line of evidence is undercut by the manner of its confirmation: testing whether specific mutations in CPSF3 cause resistance requires that the mutations be introduced WITHOUT AN3661 selection, and once the appropriate rearrangement has been confirmed, testing for resistance. All the mutants in the Palencia paper, and most of the mutants in the Sonoiki paper were isolated by selecting the transfected parasite populations with AN3661, making it tautological to later observe resistance in the resulting clones, and confounding their interpretation. In other words, the only proper experimental confirmation that mutations in CPSF3 can cause AN3661 resistance is provided by Sonoiki et al. through the generation and analysis of the transfectant population C4.

Reply: The reviewer correctly points out that most of the *Plasmodium* transfections were performed with brief application of AN3661 pressure. However, as the reviewer states, an experiment was performed with selection only for the presence of the plasmids (with WR99210 and blasticidin), without AN3661 pressure. That experiment yielded *cpsf3*-modified lines, and the observed level of AN3661 resistance was equivalent to that seen with AN3661 selection of cultured parasites. Thus, in addition to the biochemical confirmation noted above, definitive genetic confirmation of the *P. falciparum* CPSF3 target for AN3661 has been provided. Further, we point out that AN3661 resistance typically occurs in one per $\sim 10^9$ parasites. Our electroporations begin with $\sim 2 \times 10^7$ parasites, and we estimate that only ~ 80 of these would be transformed based on published assessments of transfection efficiency. Thus, quantitatively, it is highly unlikely that a mechanism independent of *cpsf3* mutation was selected by AN3661 in our transfection experiments. To more overtly make the important distinction between selection with or without AN3661, we have reworded a key sentence in Results (page 8) to: “While most transfections included selection with 170 nM AN3661, importantly transfectant C4 (PfCPSF3 Y408S) did not, and rather was obtained using only WR99210 and blasticidin selection of the editing plasmids.”

2) I therefore strongly suggest that the two studies be merged in order to proceed with publication at Nature Communications, taking into account the following reasons: (1.) identification of CPSF3 mutations by Palencia et al. rely on the prior identification of such mutations by Sonoiki et al., (2.) demonstration that the CPSF3 mutations were causal was not appropriately performed by Palencia et al., (3.) Palencia et al. provide important evidence for the function of the compound in an in vivo model of toxoplasmosis, and (4.) Sonoiki et al. provide the only analysis of the molecular role of CPSF3 on mRNA levels by performing the Northern blots presented in figure 6. The pharmacological properties of this compound, its activity at low nM concentrations, and its confirmed utility in mouse models of toxoplasmosis, highlight the importance of this work and I hope that the authors will find a way to merge the studies.

Reply: We respectfully suggest that it would be impractical to merge these two manuscripts. The *P. falciparum* story, in particular, is already large, and adding the *Toxoplasma* story will require near doubling of the size of the manuscript, as the two studies were done independently, with mostly distinct methodology. Indeed, only two senior authors are shared on the two manuscripts. Further, collapsing the reports into one manuscript would be unfair to Andres Palencia, the first author of the *Toxoplasma* report. We are aware that Dr. Palencia is responding to his review with a revised *Toxoplasma* manuscript. We hope that it will be possible to co-publish the two manuscripts, but our *P. falciparum* manuscript can stand alone.

3) The general presentation of the figures could be improved by merging Fig. 1 and Fig. 2, and including the *P. berghei* data in the main figure. I think the latter is important because it provides an in vivo validation of the compound's effect.

Reply: We respectfully do not see particular value in combining Fig. 1 (two panels) and Fig. 2 (two panels) into a single figure. We are not making this change now, but will do so if requested by the editor. Regarding inclusion of figures on murine malaria in the main text, we agree that these data will add to the report, and in fact we have taken this suggestion a step further. We added two sub-figures, the requested figure on *P. berghei* (moved from supplementary material) and a new figure on the *P. falciparum* mouse model results. These are now panels a and b of a new Fig. 2. These results demonstrate potent in vivo antimalarial activity of AN3661.

4) Exploring the correlation between the different mutations in Fig. 3 and Fig. 4 might strengthen the case for their effect, since it appears that the both spontaneous and targeted mutations of Y408 have stronger effects than mutation of other CPSF3 residues.

Reply: We appreciate the reviewer's request, but we are concerned about over-interpreting our data. We noted the following in the first version of our manuscript (page 8): "In the engineered parasites, the PfCPSF3 T406I and Y408S mutations conferred a 12 to 50-fold decrease in AN3661 sensitivity, similar to changes in sensitivity in parasites that acquired these mutations after *in vitro* resistance selection." Considering experimental noise and model uncertainty, we are not confident about highlighting relatively minor differences in mutation effects (i.e. differences amongst 12-50 fold decreases in AN3661 sensitivity), all of which are shown in Fig. 4 and 5 (formerly Fig. 3 and 4). However, in light of these comments, we have added the following sentence immediately after that copied above (page 8), which highlights the relative difference in mutation effect noted by our reviewer: "Selected and transfected parasites with the Y408S mutation were consistently less sensitive to AN3661 compared to parasites with the T406I mutation."

5) Fig. 6 would be significantly strengthened by providing quantification of the Northern blots across several experimental replicates for WT and C4 parasites.

Reply: As above we regret that we did not clearly describe replicate experiments. We do not have quantification data, but this experiment was performed 3 times, with nearly identical

results. We have added the following to the Fig. 7 (formerly Fig. 6) legend. “The figure shows results from a single experiment. Two additional experiments yielded the same results.”

REVIEWERS' COMMENTS:

Reviewer #2 (Remarks to the Author):

The authors have addressed my concerns and I have no additional comments

Reviewer #3 (Remarks to the Author):

The authors have adequately addressed my concerns from the previous review. The data presented strongly support the proposed model of AN3661 action through the inhibition of PfCPSF3.

My only lingering concern regards the current Figure 7. This is an important figure because it informs the function of PfCPSF3 and demonstrates the consequence of its inhibition on mRNA stability. If, as stated, the authors have data from three different experiments showing "the same" results, then it should be no problem to perform the requested densitometry to display the reproducibility of such an effect. Alternatively, the authors could include the other two experiments as supplemental figures. Simply stating that the experiment is reproducible, without evidence, is not appropriate for publication.

Thank you for re-review of our manuscript (NCOMMS-16-21360A: “A potent antimalarial benzoxaborole targets a *Plasmodium falciparum* cleavage and polyadenylation specificity factor homologue”). The manuscript was judged acceptable for publication except for one additional concern expressed by Reviewer #3. We have copied the reviewer comment in full below, followed by our response. In addition, we received a copy-edited version of our manuscript. We have addressed all requests and submit a revision with changes shown using Track Changes. We are happy to comply with the *Nature Communications* transparent review system. The draft summary to accompany our manuscript is acceptable to us.

Reviewer #3 (Remarks to the Author):

The authors have adequately addressed my concerns from the previous review. The data presented strongly support the proposed model of AN3661 action through the inhibition of PfCPSF3. My only lingering concern regards the current Figure 7. This is an important figure because it informs the function of PfCPSF3 and demonstrates the consequence of its inhibition on mRNA stability. If, as stated, the authors have data from three different experiments showing "the same" results, then it should be no problem to perform the requested densitometry to display the reproducibility of such an effect. Alternatively, the authors could include the other two experiments as supplemental figures. Simply stating that the experiment is reproducible, without evidence, is not appropriate for publication.

RESPONSE: We appreciate the concern that reproducibility is essential. We now include Supplementary Figure 2, which includes multiple duplicate results. Consistently, incubation of WT parasites with our test compound AN3661 led to unstable messages for the 3 studied genes, but the messages were stable (prominent bands on Northern blots) in parasites selected or engineered for AN3661 resistance. We also include Supplementary Figure 3, which displays uncropped versions of all original gels used to create Figure 7.

Also, we have reviewed the edited versions of our manuscript and supplemental information. We have made minor adjustments to the edits, and also added some minor corrections in nomenclature for our description of experiments using the CRISPR system, with all changes shown with Track Changes.

Please let us know if any additional changes are needed for our manuscript.

Sincerely,

Philip J. Rosenthal